# GEPCode: A Context-Aware 1M-Parameters Graph-Based Language Model for Source Code

## Abstract

The pursuit of optimal conditions for software execution poses a complex challenge. This task can be automated by harnessing the structured nature of programming languages, especially from compiler intermediate representations of code (IR). The manipulation of source code using Large Language Models (LLMs) is a thriving area of study in Natural Language Processing (NLP) literature. However, in this study we illustrate how we can circumvent the need for exceedingly large models by employing domain-specific language models. These models have a reduced number of parameters but retain the ability to capture the relationships within source code elements. We introduce GEPCode, a graph neural network designed to model IR with the flexibility to adapt to new tasks. This flexibility is obtained through special "meta" nodes, that allow for the representation of additional task-dependent contextual information. Pre-training is performed by solving node and graph-level tasks, resulting in a general language model. After a fine-tuning phase on two downstream tasks, Device Mapping and Algorithm Classification, we achieve average accuracy results of 88.9% (NVIDIA) and 92.3% (AMD) for the former and 97.2% for the latter. Comparing our methodology with state-of-the-art models trained from scratch, our results are similar or better, yet providing a more flexible model. Moreover, we achieve similar accuracy results in downstream tasks compared to state-of-the-art pre-trained language models based on Transformers, while utilizing 100 times fewer parameters.

## 1 Introduction

The current landscape of computing systems is characterized by high complexity in hardware architectures and configurations, as well as in programming languages, techniques, and compilation options. Achieving optimal software execution performance often requires thorough exploration of various configuration parameters and manual profiling of source code across different compute units. However, this process becomes impractical as the number of possible alternatives increases(Magni et al., 2014; Ivanov et al., 2024). Recently, deep learning techniques based on Natural Language Processing (NLP), such as Language Models (LMs), have been utilized to address this complexity (Zhang et al., 2024; Allamanis et al., 2018). Current research focuses on two main approaches: custom end-to-end architectures, that are trained from scratch on a single task, and more general Language Models (LMs), that are pre-trained on a large amount of code samples and can be fine-tuned on a variety of downstream tasks. The key question in this context is whether alternative representations of code can be used to develop general, efficient, and compact language models of source code. Answering this question could help bridging the gap between the efficiency of task-specific architectures and the generality of larger language models.

We present GEPCode, a Graph-based, Efficient, Pre-trained, Context-aware Language Model (LM) of graph representations of source code. GEPCode leverages a graph-based representation of code, following recent works highlighting their ability to capture structural patterns related to the causal and temporal dependencies between data (e.g. variables and constants) and instructions (Brauckmann et al., 2020; Cummins et al., 2021b; TehraniJamsaz et al., 2023; Yamaguchi et al., 2014; Guo et al., 2021). GEPCode tackles several open problems in the field. Firstly, many optimization-related tasks require considering additional task-dependent contextual information. For instance, the task of heterogeneous device mapping (i.e. predicting which device would run a given program faster in a heterogeneous machine) pairs code samples with dynamic parameters affecting decisions, such as the

size of input data. Previous solutions often separate the encoding of the code sample from that of the external parameters (Cummins et al., 2021b; 2017a; Ben-Nun et al., 2018; Barchi et al., 2019; 2021; Parisi et al., 2022; Brauckmann et al., 2020; Hakimi et al., 2023). However, we argue that it would be better to insert this contextual information inside the representation, allowing models to reason upon it during processing, constructing context-aware source code encodings. Then, we address the size of recent pre-trained models of source code (Niu et al., 2023; Feng et al., 2020; Guo et al., 2021; Wang et al., 2021; Peng et al., 2021). While large-scale models achieve state-of-the-art performance across various downstream tasks, we posit that efficiency should be prioritized in the context of source code optimization, especially where computing and memory resources may be limited. We also consider recent studies that are critical of the effectiveness of Large Language Models (LLMs) for source code optimization and analysis (Chen et al., 2023; Fang et al., 2024; Karmakar & Robbes, 2021). In this work, we show that our model achieves comparable results to those of Transformer-based LMs while using over 100 times fewer parameters. This positions GEPCode as competitive alternative, and will hopefully encourage a discussion on the trade-offs within this domain. To achieve these results, we developed an effective pre-training pipeline that incorporates both graph-level and node-level targets for enhanced robustness. We pre-train GEPCode on a large collection of code (Armengol-Estapé et al., 2022), obtaining a general and flexible model.

Our contribution can be summarized as follows: i) We design a novel graph-based source code representation introducing an innovative method for incorporating contextual information directly into model reasoning; ii) We develop a language model (LM) for our representation by pre-training a Graph Neural Network (GNN) on a large and diverse dataset of source code samples through a novel technique that fully leverages the information in our representation; iii) We evaluate our model across various downstream tasks, demonstrating the capability of our model to compute effective representations of source code. We show that our results are comparable or even better than those of larger pre-trained models, despite containing over 100 times fewer parameters. We achieve an average accuracy of 90.6% on the heterogeneous device mapping task, and of 97.2% on algorithm classification.

The rest of this paper is organized as follows. Section 2 presents related works that address similar problems in literature. Section 3 details our novel graph-based source code representation. Section 4 describes the architecture of our language model and the pre-training and fine-tuning tasks. Section 5 reports experimental results and compares our model to the existing literature. We also propose ablation studies to motivate our main design choices. Finally, Section 6 wraps up the work.

## 2 BACKGROUND AND RELATED WORKS

Several works (Cummins et al., 2017a; Vavaroutsos et al., 2022) have employed Recurrent Neural Networks (RNN), especially Long Short-Term Memory (LSTM) cells (Hochreiter et al., 1997), as the core mechanism for operating on sequences of raw code tokens. In order to benefit from structural features of code and to create LMs that are independent from specific source languages, other studies (Barchi et al., 2019; 2021; Ben-Nun et al., 2018; Brauckmann et al., 2020; VenkataKeerthy et al., 2020; Hakimi et al., 2023; Niu et al., 2023) have explored the option to work on tokens of LLVM-IR code, a low-level IR used internally by the LLVM compiler (Lattner et al., 2004) which offers explicit memory-related operations and facilitates access to control and data flow. Graph representations are also common transformations in the compilation pipeline and can easily be extracted from IR. For instance, Abstract Syntax Trees (ASTs), depicting the syntactic structure of source code, Control Flow Graphs (CFGs) and Data Flow Graphs (DFGs), representing code operations by means of dependencies between data and instructions, are directly employed or combined with other inputs by some source code language models (Brauckmann et al., 2020; Ben-Nun et al., 2018). More recently, works such as ProGraML (Cummins et al., 2021b) and its extension Perfograph (TehraniJamsaz et al., 2023) have designed expressive graph-based representations that can be easily employed in Deep Learning pipelines.

Transformer-based models (Vaswani et al., 2017) have recently emerged for graph modeling (Ying et al., 2021; Dwivedi & Bresson, 2021; Zhang et al., 2020; Shirzad et al., 2023). However, these models often have billions of parameters, making their training and inference processes resource-intensive. Therefore, we focus on pre-training methods specifically designed for GNNs, which typically have fewer parameters. Masked Graph Autoencoders (Li et al., 2023a; Hou et al., 2022;

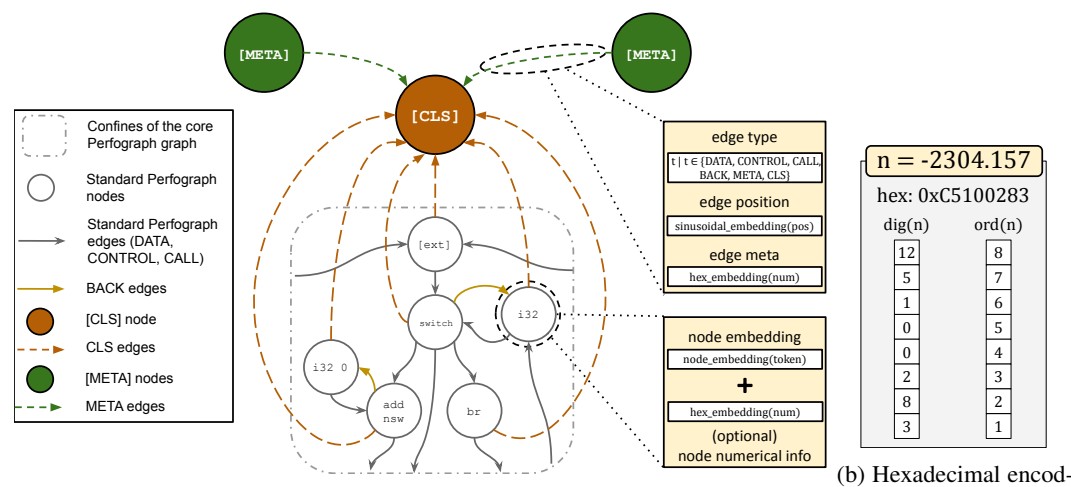

(a) Example detailing the main components of our source code representation.

(b) Hexadecimal encoding for a number $n$.

Figure 1: The main contributions in our graph representation of source code.

2023; Tu et al., 2023; Tian et al., 2023) and Contrastive self-supervised learning (Wu et al., 2021; Xia et al., 2022) are widely used frameworks in this context. The former masks elements of the input graphs and uses an encoder-decoder architecture to reconstruct the original elements, while the latter works by generating multiple "views" for each graph through data augmentation and by training models to maximize an agreement measure within views of the same graph. In this work, we employ both techniques to pre-train our network.

Existing literature on graph pre-training methodologies acknowledges that discrepancies between pre-training and fine-tuning tasks often result in decreased performance on downstream tasks (Lu et al., 2021; Liu et al., 2023a; Sun et al., 2023; Li et al., 2023b; Liu et al., 2023b; Wang et al., 2024). On the other hand, many current graph-based representations of source code (Cummins et al., 2021b; Brauckmann et al., 2020; Ben-Nun et al., 2018; Yamaguchi et al., 2014) frame program-level tasks as graph-level problems, while several pre-training techniques focus primarily on node-level or edge-level targets without further addressing this gap (Guo et al., 2021; Zhang et al., 2020; Hu et al., 2020b; Tu et al., 2023; Li et al., 2023a; Hou et al., 2022; 2023; Tian et al., 2023). Instead, our representation design allows to cast program-level tasks as node-level problems, bridging the divide between node-level pre-training and downstream tasks.

## 3 SOURCE CODE REPRESENTATION

The proposed graph-based representation expands upon the approach introduced in ProGraML (Cummins et al., 2021b) and further extended in Perfograph (TehraniJamsaz et al., 2023). Our representation expresses LLVM-IR code samples as graphs $G = (V, E)$, where $V$ is the set of nodes and $E$ is the set of edges. Fig. 1a shows a schematic example of the representation. Each node $v \in V$ is mapped to a token within a vocabulary of LLVM-IR elements, comprising instruction names (e.g. add, switch, br, ...), data types (e.g. i32, <2 x double>, ...), and so on. Hard-coded constants may be annotated with the value of the variables they represent (e.g. i32 0), while all external dependencies are represented by a single [ext] node. Edges are directed and represent dependencies between the elements of code. They have a *type* attribute, specifying the kind of dependency among DATA (e.g. an instruction using or returning a variable), CONTROL (e.g. an instruction following another) or CALL (e.g. an instruction calling a function, or a function returning a value to the caller). Edges also have a *position* attribute, distinguishing operands order. In the following sections, we provide a detailed description of our source code representation extensions.

### 3.1 NUMERICAL ENCODINGS

We propose a novel method for embedding numerical values, enriching the node representation of hard-coded constants and variables, and the edge representation of contextual dependencies. Perfograph (TehraniJamsaz et al., 2023) implements a similar concept, but their approach does not

differentiate between positive and negative numbers and necessitates preliminary processing steps to handle the large diversity of digit counts. In contrast, our method employs a signed, fixed-size numerical representation that can be easily vectorized and processed in parallel with the rest of the data. Our representation encodes numbers using two 8-dimensional vectors (*digit* and *order*), as can be seen in Fig. 1b. All numbers are explicitly transformed into single-precision floating points, then converted into their hexadecimal representation through the IEEE-754 encoding standard. The *digit* vector is populated by the 8 hexadecimal digits, mapped to the range 0-15 for convenience, while the *order* vector contains the values of the 8-1 range indicating their order. Since the IEEE-754 standard encodes sign into the first bit, the sign in our representation is implicitly present into the first digit.

## 3.2 Node-level Description

Our new source code representation extensions, aim to achieve two primary goals: aggregating a global graph representation into a specialized node in order to bridge the gap between node-level pre-training and program-level fine-tuning tasks, and integrating contextual information into the graph representation. Contextual information can include simple graph properties, such as the diameter of the graph, but also key features for guiding decisions in downstream tasks, like the size of an input matrix for a kernel or device and framework parameters. To this end, we include two novel node types: i) `[CLS]`, collecting a global graph representation; ii) `[META]`, representing general contextual meta-information related to the code sample or the graph. Each graph contains a single `[CLS]` node and a variable number of `[META]` nodes, depending on the availability of external information for the task. Nodes are mapped to feature vectors on the basis of a vocabulary $K$ comprising the most frequent $|K| = 344$ tokens extracted from a large dataset of LLVM-IR code files compiled from various open-source projects. Note that all `[META]` nodes are mapped to the same feature vector, representing "general contextual information"; actual meta-information, are instead encoded in edges (see Section 3.3). For a detailed analysis of the dataset and of the vocabulary extraction process, please refer to Appendix A.

## 3.3 Edge-level Description

In the graphs collected for our vocabulary-creation step, we observed that 6% of nodes lack incoming connections, typically indicating variables and constants that are not outputs of prior operations but are used by later instructions. Therefore, we introduce a new edge type, `BACK`, to connect `DATA` dependencies back to their sources, improving the connectivity within the graphs. Additionally, we propose two novel edge *types*: i) `META`, connecting `[META]` nodes to a `[CLS]` node. They allow `[CLS]` nodes to receive meta-information, enabling a more specialized global graph representation. We note that these connections are unidirectional, so that the `[CLS]` node only acts as a receiver and has no outgoing connections, preserving the original graph structure; ii) `CLS`, connecting non-`[META]` and non-`[CLS]` nodes to a `[CLS]` node. They enable the `[CLS]` nodes to receive and aggregate information from all other nodes within the graph, allowing the iterative construction of the global representation (`[CLS]` and `[META]` refer to node tokens when enclosed in square brackets, to edge types otherwise). Moreover, we incorporate *meta-information* as additional features included in edges. These features are only significant in `META` edges; for other edge types, they are replaced by zero-padding. Since the nature of available meta-information varies depending on the application, we employ the previously described numerical encodings in order to incorporate diverse information avoiding type limitations.

# 4 Language Modeling

In this Section, we report the process behind our pre-training and fine-tuning experiments, describing the model architecture, as well as the employed datasets and training tasks.

## 4.1 Model Architecture

Initially, every node $v \in V$ is mapped to a learnable feature vector $h_v^{0*}$ by lookup in a fixed-size embedding table $E_v \in \mathbb{R}^{|K| \times d}$, where $d$ denotes the hidden dimensionality of the network. Numerical encodings for numbers contained within nodes are processed to generate fixed-size embeddings:

$$h_n = (E_{\text{dig}}(\text{dig}(n)) + E_{\text{ord}}(\text{ord}(n))) \tag{1}$$

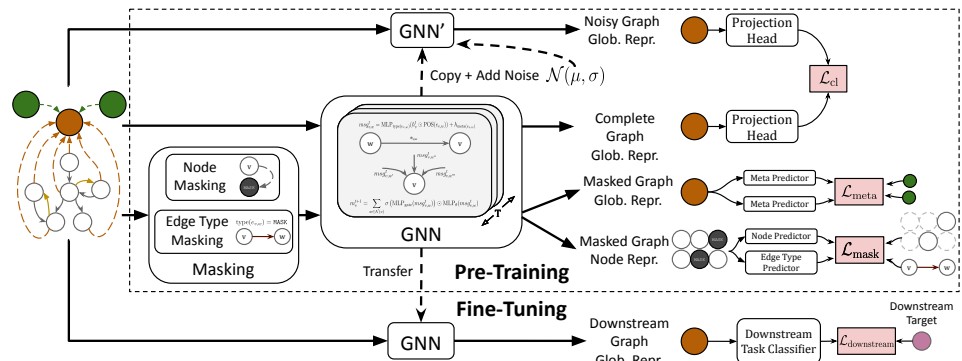

Figure 2: Scheme for the pre-training and fine-tuning phases.

where $E_{(*)}$ are specialized embedding tables for the digits (retrieved by function "dig") and orders (retrieved by function "ord"). The initial value of each node $h_v^0$ is then computed by summing $h_v^{0*}$ and its optional numerical embedding together.

We implement a GNN-based LM processing the input graphs through a local aggregation mechanism called *message passing*, repeated for a number of steps $T$. At each step, $t \in [0, \ldots, T-1]$, three fundamental operations are executed: i) *Message emission*: A message for each pair of neighboring nodes $(v, w)$ is generated by modulating the source features $h_v^t$ by their edge *position* attribute and processing the result through a specific MLP for the connection *type*. Meta-information are embedded as in Eq. 1 and added to the messages.

$$msg_{v,w}^t = \text{MLP}_{\text{type}(e_{v,w})}(h_v^t \odot \text{POS}(e_{v,w})) + h_{\text{meta}(e_{v,w})} \tag{2}$$

where $\odot$ denotes the Hadamard product and $\text{POS}(e_{v,w})$ is implemented as a sinusoidal encoding (Vaswani et al., 2017). ii) *Message aggregation*: Nodes receive messages from each incoming connection and aggregate them through an attention-based mechanism:

$$m_v^{t+1} = \sum_{w \in \mathcal{N}(v)} \sigma\left(\text{MLP}_{\text{gate}}(msg_{v,w}^t)\right) \odot \text{MLP}_\theta(msg_{v,w}^t) \tag{3}$$

where $\sigma$ is the sigmoid function, $\text{MLP}_{\text{gate}}$ maps messages to an attention score and $\mathcal{N}(v)$ is a function returning the neighbors of node $v$. iii) *Update*: Function $U$, a Gated Recurrent Unit (GRU) cell (Cho et al., 2014), updates nodes features based on their current state and the aggregated message $m_v^{t+1}$:

$$h_v^{t+1} = U\left(h_v^t, m_v^{t+1}\right) \tag{4}$$

At the end of message passing, node representations contain contextual information for their $T$-step neighborhood, while the [CLS] node holds the global, meta-informed graph representation.

## 4.2 PRE-TRAINING

During pre-training, we expose our GNN to a vast and diverse collection of graph representations of source code. We pre-train our model utilizing the *synth-compilable* subset of the Exebench dataset (Armengol-Estapé et al., 2022). We employed Clang (Clang) for compiling code into LLVM-IR with -O1 compilation level, then used a custom version of the ProGraML's Python library (Cummins et al., 2021b) to turn source code into a graph representation and applied Perfograph and our source code representation extensions directly at this level. Only 70% of the available samples, equivalent to 1.6 M, were successfully processed into our representation. Our training set comprises 1.3 M graphs, with the remaining graphs are equally split between validation and testing.

### 4.2.1 TASKS DEFINITION

Our representation models global graph information as node features into the [CLS] node. Graph-level tasks are therefore easily expressible in terms of node-level tasks, while edge-level tasks (such as link prediction) can also be cast as node-level problems by aggregating the two ends of a connection into a single representation (e.g. by feature-wise product or sum). In other words, models using our representation can be robustly pre-trained simply using node-level self-supervision. We propose to solve three tasks in parallel: *Attribute Masking*, *Meta Prediction* and *Contrastive Learning*. Fig. 2 exemplifies the usage of the model during the pre-training and fine-tuning phases.

**Attribute Masking**  The Attribute Masking task (Hu et al., 2020a) is akin to Masked Language Modeling (MLM) in BERT (Devlin et al., 2018). We mask a random subset of nodes and edge types within the graph, replacing node encodings with a special `[MASK]` token and edge types with a special `MASK` type with probability $p$. During the random sampling process, we deliberately avoid masking the `[CLS]` and `[META]` nodes and `CLS`, `MASK` and `BACK` edges, so that only the nodes and edges of the core graph are affected. At the end of the $T$ GNN message passing steps, we employ two separate learnable linear projections $D_v \in \mathbb{R}^{d \times |K|}$ and $D_e \in \mathbb{R}^{d \times 3}$ (where 3 is the size of the set of maskable edge types, `DATA`, `CONTROL`, `CALL`) to compute probability distributions over the target spaces for each masked attribute. The loss for this task $\mathcal{L}_m$ is a sum of two categorical cross-entropy terms ($\sigma$ denotes the softmax function).

$$\mathcal{L}_{\text{mask}} = \frac{1}{|V_{\text{m}}|} \sum_{v \in V_{\text{m}}} -\log\left(k_v^* \cdot \sigma\left(D_v^\top h_v^T\right)\right) + \frac{1}{|E_{\text{m}}|} \sum_{(v,w) \in E_{\text{m}}} -\log\left(\text{type}^*(e_{v,w}) \cdot \sigma\left(D_e^\top h_v^T\right)\right) \quad (5)$$

where $V_m$ and $E_m$ represent the two sets of masked nodes and edges, $h_v^* \in \mathbb{R}^{1 \times |K|}$ and $\text{type}^*(e_{v,w}) \in \mathbb{R}^{1 \times 3}$ are the one-hot encoded targets (original node token and edge type).

**Meta Prediction**  To enhance the network's ability to effectively capture meta-information into the global representation, we also task the model with predicting 3 graph properties i) graph diameter; ii) average node degree; iii) graph clustering coefficient. We pre-computed these statistics on the training sets of Exebench (Armengol-Estapé et al., 2022) in order to analyze their distribution and determine suitable thresholds for normalization. Subsequently, we introduced a `[META]` node for each property into all graphs, connecting them to the `[CLS]` node accordingly. The respective `META` edges contain the numerical (or hexadecimal) encoding of the normalized property value. Following the message passing phase, we map the final representation of the `[CLS]` node $h_G^T$ to predictions using a distinct linear layer $\text{MLP}_m$ for each property $m$. The loss function for this task $\mathcal{L}_M$ is the average mean squared error (MSE).

$$\mathcal{L}_{\text{meta}} = \frac{1}{|M|} \sum_{m \in M} \left(m - \text{MLP}_m\left(h_G^T\right)\right)^2 \quad (6)$$

**Contrastive Learning**  Finally, we propose incorporating a graph-level task to increase the robustness of the global representation. We adapt SimGRACE (Xia et al., 2022), a graph contrastive learning technique designed to maximize the agreement between different representations of the same graph, while distancing the representations of distinct graphs. Unlike traditional contrastive learning approaches that compute a secondary representation of inputs using data augmentations, SimGRACE generates an alternative view $h_G^{T'}$ for input graph $G$ by perturbing the model's parameters with Gaussian noise and passing the input through the network a second time. Gradients are not computed during this second pass. Subsequently, a projection head is applied to the final representations, which are compared using cosine similarity (denoted as sim) against the representations of other graphs in the mini-batch $B$. A temperature hyper-parameter $\eta$ is employed to increase label entropy. The loss for this task $\mathcal{L}_{\text{cl}}$ is thus defined as:

$$\mathcal{L}_{\text{cl}} = \frac{1}{|B|} \sum_{G \in B} -\log \frac{\exp\left(\text{sim}\left(h_G^T, h_G^{T'}\right)/\eta\right)}{\sum_{g \in B, g \neq G} \exp\left(\text{sim}\left(h_G^T, h_g^T\right)/\eta\right)} \quad (7)$$

### 4.2.2  PRE-TRAINING PIPELINE

Computing the overall loss necessitates careful definition of operation flow. Contrastive Learning requires no masking during model processing, as the source of variability between representations is the perturbation of parameters. Conversely, the Attribute Masking task demands that the GNN processes a partially masked graph, while there are no special requirements for Meta Prediction. To combine these requirements into a unified pipeline, we use 3 separate GNN passes. Given a mini-batch of graphs $B = [G_1, \ldots, G_N]$, we first pass the unmasked graphs into the network, computing $h_G^T$ for each $G \in B$ in Eq. 7. Then, we block the gradient and perturb the network's parameter using Gaussian noise. We pass the unmasked graphs into the network a second time, computing $h_G^{T'}$ for each $G \in B$ in Eq. 7. Finally, we restore gradient computation and the original model parameters, as we mask the input graphs and pass them to the GNN a third time. This step computes $h_v^T$ and $h_G^T$

for all involved nodes and graphs in Eq. 5 and 6. A final loss $\mathcal{L}$ is calculated as a weighted sum of the various elements, where the coefficients have been selected to normalize the range of each loss function (in this work, $\lambda_{\text{mask}} = 1$, $\lambda_{\text{meta}} = 2$ and $\lambda_{\text{cl}} = 0.5$). This ensures that each loss contributes equally to the overall optimization process.

$$\mathcal{L} = \lambda_{\text{mask}}\mathcal{L}_{\text{mask}} + \lambda_{\text{meta}}\mathcal{L}_{\text{meta}} + \lambda_{\text{cl}}\mathcal{L}_{\text{cl}} \tag{8}$$

### 4.3 DOWNSTREAM TASKS

We evaluate GEPCode on two downstream tasks aimed at optimizing compile-time choices and testing the representation capabilities of the model: *Heterogeneous Device Mapping* (*DevMap*) and *Algorithm Classification*. For these tasks, we transform the available source code into our graph-based representation following the procedure designed for the pre-training dataset, compiling code with the `-O1` optimization level in order to maintain a similar input distribution and inserting `[META]` nodes as appropriate. The graph is then processed by our LM, initialized using the weights obtained at the end of the best pre-training epoch in terms of validation loss. A final MLP classifier is appended at the end of the model in order to map the produced representations to the decision space according to the task. All reported results are averaged over 5 experiments with different random seeds.

#### 4.3.1 HETEROGENEOUS DEVICE MAPPING

The DevMap task concerns predicting the most efficient device for executing a kernel. This is a crucial task in the context of embedded systems, where a vast heterogeneity of hardware configurations exists. Results for this task are assessed using the DevMap dataset, introduced in (Cummins et al., 2017a;b). This collection contains 680 samples of OpenCL kernels, each paired with two auxiliary values: the Work Group size, affecting the amount of parallelism, and the size of input data, affecting transfer time between host and executing device. Each combination has been run on the CPU and GPU of 2 separate heterogeneous machines, resulting in two distinct versions of the dataset (NVIDIA and AMD, depending on the GPU model). Before transforming code into our graph representation, we re-introduce external imports and constants into the raw kernels. Auxiliary inputs are represented as `[META]` nodes, and their numerical information is normalized and inserted into the respective edges. The dataset is notoriously small and unbalanced; specifically, the DevMap NVIDIA dataset has a distribution of 43% CPU and 57% GPU, while the DevMap AMD dataset shows a distribution of 58% CPU and 42% GPU. Therefore, we employ *stratified 10-fold cross-validation* for training and evaluating the model, reporting the Matthews Correlation Coefficient and F1 Score, as proposed in (Parisi et al., 2022).

#### 4.3.2 ALGORITHM CLASSIFICATION

A general language model of code should be able to recognize high-level features that are resistant to minor variations in implementations. To this end, the objective of Algorithm Classification is to categorize programs based on the problems they address. For this task we employ POJ-104 (Mou et al., 2015), a collection of 104 classes of algorithms, each exemplified by about 500 C++ programs. We split the dataset by randomly sampling 80% of the code samples for the train set (about 300 programs per algorithm) and evenly distributing the remaining files between validation and testing. For this task there are no meta-inputs, so no `[META]` node is added to the graphs. The resulting representation is processed through the GNN, and a final a 104-way classifier selects the appropriate algorithm class.

## 5 EXPERIMENTAL RESULTS AND ANALYSIS

In this Section, we report the results of our experiments. We compare GEPCode with previous works in literature, and we perform several ablation studies aimed at motivating our design choices.

### 5.1 SETUP

We implemented our models and data processing procedures in PyTorch and PyTorch Geometric (Paszke et al., 2019; Fey et al., 2019). For all experiments, we use $T = 6$ steps of message passing, a hidden dimensionality and initial embedding size of 256 and an Adam optimizer (Kingma & Ba,

Table 1: Result comparison. Pre-training dataset, approximate number of parameters and standard deviation are reported when disclosed or applicable.

| Model Name | Pre-train set | Core Arch. | DevMap Accuracy | | POJ-104 Accuracy | Params. ($\times 10^6$) |
| --- | --- | --- | --- | --- | --- | --- |
| | | | NVIDIA | AMD | | |
| ProGraML Cummins et al. (2021b) | - | GNN | .800 | .866 | .962 | 0.09 |
| DeepTune Cummins et al. (2017a) | - | RNN | .803 | .837 | - | 0.08 |
| CDFG Brauckmann et al. (2020) | - | GNN | .814 | .864 | - | 0.09 |
| DeepTune Exp. Vavaroutsos et al. (2022) | - | RNN | .815 | .874 | - | - |
| DeepLLVM Barchi et al. (2019) | - | RNN | .823 | .853 | - | 0.08 |
| DeepLLVM-CNN Barchi et al. (2021) | - | CNN | .873±.009 | .890±.006 | - | 0.08 |
| IR2Vec VenkataKeerthy et al. (2020) | - | No-DL | .887 | .913 | .961 | - |
| Siamese DeepLLVM Parisi et al. (2022) | - | CNN | .888±.009 | .917±.007 | - | 0.08 |
| Perfograph TehraniJamsaz et al. (2023) | - | GNN+Manual | .900 | .940 | .950 | 0.05 |
| DeepLLVM-CNN+ML Hakimi et al. (2023) | - | CNN+ML | .911 | .922 | .955 | - |
| Inst2vec Ben-Nun et al. (2018) | NCC | SkipGram+RNN | .820 | .828 | .948 | 0.6 |
| CodeBERT Feng et al. (2020) | CodeSearchNet Husain et al. (2019) | Transformer | .868 | .956 | .954 | 125 |
| CodeT5 Wang et al. (2021) | CodeSearchNet | Transformer | .885 | .931 | .959 | 220 |
| IRGen Li et al. (2022) | POJ104/GCJ | Genetic+CNN | .899 | .943 | .980 | - |
| OSCAR Peng et al. (2021) | OSCAR | Transformer | .895 | .941 | .981 | 163 |
| FAIR Niu et al. (2023) | OSCAR | Transformer | .916 | .965 | .983 | 138 |
| GEPCode-100k | Exebench Armengol-Estapé et al. (2022) | GNN | .852±.012 | .911±.003 | .955±.006 | 0.13 |
| **GEPCode** | Exebench | GNN | .889±.008 | .923±.008 | .972±.001 | 1.3 |

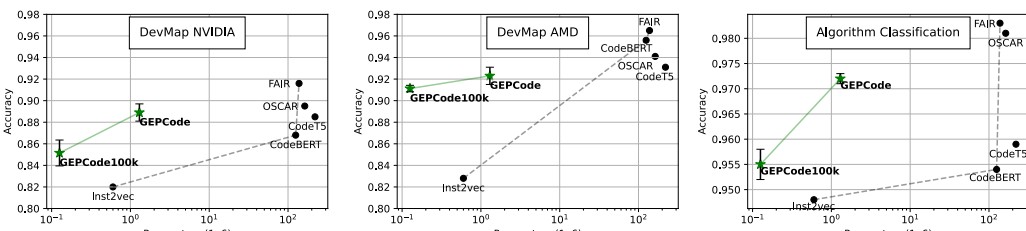

Figure 3: Model accuracies with respect to the number of parameters. Models lacking either the number of parameters or accuracy results are not shown. We also indicate the Pareto front of previous models.

2017). We also designed a small-scale version of GEPCode (*GEPCode100k*) that only uses 125 k parameters by reducing the amount of message passing layers ($T = 4$) and using a dimensionality of 64. By default, we use an Adam optimizer with a learning rate of $2.5 \times 10^{-4}$ and a dropout rate of 0.3. We pre-train until convergence with a fixed mask rate of 0.4, using a batch size of 64. Most of our pre-training experiments lasted ∼30-40 hours on a single Quadro RTX 6000 GPU, comprising 65-75 k training steps. Fine-tuning for both downstream tasks runs for 100 epochs. The final classifiers for the tasks are 2-layer MLPs with sizes [64, $out$], where $out = 2$ for DevMap and $out = 104$ for Algorithm Classification.

## 5.2 RESULTS

On the DevMap task, GEPCode achieves an accuracy of 88.9% on the NVIDIA variant of the dataset, and of 92.3% on the AMD variant, with a standard deviation of 0.8%. The Matthews Correlation Coefficient (MCC) is 0.775 and 0.854 and the F1-scores are 0.902 and 0.914 respectively. The predictions of the model lead to a 1.45x speedup on the NVIDIA dataset, where predictions are compared against the naive choice of always running kernels on GPU, and to a 3.34x speedup on the AMD dataset, where we use CPU times as baseline since they frequently outmatch GPU times on this variant. For the task of Algorithm Classification, we instead achieve a test accuracy of 97.2%.

We compare the results of our methodology to both end-to-end approaches and pre-trained LMs of code in Table 1 and visually in Fig. 3. Our model exhibits a minor performance drop on the DevMap and Algorithm Classification tasks with respect to other pre-trained Transformer-based models, but it achieves a considerable gain in terms of efficiency, with two orders of magnitude fewer parameters, and outperforms pre-trained models with a comparable number of parameters. We further observe that, while end-to-end solutions that achieve better results on DevMap exist, they lack generality, resulting in a considerably inferior performance on Algorithm Classification. Finally, we observe that the performance of GEPCode100k is still remarkable, considering the reduced number of parameters.

Table 2: Ablation studies results.

| Experiment name | DevMap Accuracy | |
| --- | --- | --- |
| | NVIDIA | AMD |
| baseline | $.8594 \pm .0090$ | $.8735 \pm .0110$ |
| aggr-concat-hex | $.8541 \pm .0097$ | $.8809 \pm .0142$ |
| CLS-concat-hex | $.8671 \pm .0084$ | $.8768 \pm .0118$ |
| CLS-META-hex | $\mathbf{.8891 \pm .0069}$ | $\mathbf{.9285 \pm .0055}$ |

Our experimental setup includes a server equipped with an Intel Xeon 5220 CPU (72 cores) and an Nvidia Quadro RTX 6000 GPU. The average inference time is 23 milliseconds when utilizing the GPU, compared to 239 milliseconds when using the CPU. Inference tests with CodeT5 on our setup yielded results of 112.0 ms $\pm$ 0.3 μs on GPU and 557 ms $\pm$ 74 ms on CPU.

Overall, GEPCode places itself as a good trade-off between efficiency and effectiveness. We report mean and standard deviation only for papers that include repeated experiments, ensuring a more accurate comparison.

## 5.3 ANALYSIS

We empirically evaluate the design choices presented in the previous sections through additional experiments. For all experiments, we first pre-train our GNN with the appropriate modifications and then fine-tune the weights following the same setup of Section 5.1. We start from a **baseline** that does not use [CLS] nor [META] nodes, instead concatenating normalized auxiliary inputs to a final aggregation of all node representations after $T$ message passing steps. In this experiment, we also employ numerical encodings similar to those proposed by Perfograph (TehraniJamsaz et al., 2023). Starting from this baseline, we gradually introduce the novel elements of our methodology: i) **aggr-concat-hex** uses the hexadecimal numerical representation of Section 3.1; ii) **CLS-concat-hex** adds the [CLS] node into the representation, collecting a context-independent global graph representation. Auxiliary inputs are still included by concatenation at the end of message passing and don't influence the graph representation directly; iii) **CLS-META-hex** introduces [META] nodes into the representation, allowing the creation of context-aware representations. Table 2 shows the impact of this sequence of experiments on DevMap test accuracy.

We don't observe definitive improvements from switching the baseline numerical encodings with hexadecimal representations. However, our representation is compact and efficient, using only two 8-dimensional vectors to represent any single-precision floating point number in the range $\pm \sim 3.4 \times 10^{38}$ with an exact precision of up to 7 decimal digits. The size of Perfograph numerical embeddings is instead variable and requires up to 5x larger vectors to represent a similar range.

All experiments, including baseline, aggr-concat-hex, CLS-concat-hex, and CLS-META-hex, incorporate contextual information. The key distinction between CLS-concat-hex and CLS-META-hex is that the former concatenates contextual information to the final graph representation, while the latter handles contextual information through META nodes. This approach integrates the contextual data more effectively, rather than relegating its analysis solely to final layers.

Including the [CLS] and [META] nodes into our representation has instead a clear positive effect. A T-test between the results of the **aggr-concat-hex** and the **CLS-concat-hex** experiments reveals that the statistical significance of the observed difference might be small, with p-values of 0.078 and 0.67 on the NVIDIA and AMD variants respectively. However, the difference between **CLS-concat-hex** and **CLS-META-hex** is statistically significant, with p-values smaller than 1%, motivating the inclusion of both communication mechanisms at the same time.

**Limitations** We acknowledge that our system has one main limitation: it needs compilable source code in order to generate the graph-based representation. Furthermore, large source code files could impact the memory requirements of the model, as this would result in more nodes, messages and updates throughout the message passing phase.

## 6 CONCLUSIONS

In this paper we presented GEPCode, an efficient, graph-based language model of source code that leverages graph representations to effectively capture the structural patterns of IR. We design two components that expand upon previous representations: the `[CLS]` node aggregates global features through a specialized network of connections, while `[META]` nodes represent external contextual information and allow the network to produce specialized program embeddings. We also propose a compact encoding that can be employed to process numerical information efficiently. This representation facilitates the pre-training of our LM, allowing the utilization of both node-level and graph-level tasks and reducing the discrepancies between the pre-training and fine-tuning phases. Experimental results demonstrate that our LM is able to bridge the gap between the efficiency of task-specific architectures and the generality of larger LMs, while using a limited number of parameters. For future works, we are planning to test our model on a greater number of downstream tasks and to study the impact of input graphs dimension.

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

Table 3: DCG Dataset description, indicating subset sources and the number of source code, LLVM-IR and graph files.

| Name | Cit. | License | C Files | Compiled | Graphs |
|------|------|---------|---------|----------|--------|
| Blas | Blas | BSD 3-Clause | 300 | 300 | 216 |
| bowtie2 | Langmead et al. (2012) | GPL-3.0 | 57 | 57 | 25 |
| bwa-mem | Vasimuddin et al. (2019) | MIT | 24 | 24 | 15 |
| cBench | Fursin (2014) | LGPL 2.1 | 711 | 711 | 66 |
| CLGen | Cummins et al. (2017b) | MIT | 996 | 996 | 996 |
| eigen | Guennebaud et al. (2010) | BSD 3-Clause | 4,998 | 4,998 | 3,368 |
| gemm_synth | Ben-Nun et al. (2018) | BSD 3-Clause | 3,700 | 3,700 | 3,072 |
| Gromacs | Berendsen et al. (1995) | LGPL-2.1 | 1,249 | 1,205 | 828 |
| JotaiBench | Kind et al. (2022) | GPL-3.0 | 5,535 | 5,535 | 5,535 |
| Linux | Linux | GPL-2.0 | 13,920 | 13,920 | 8,585 |
| LLVM | Lattner et al. (2004) | Apache-2.0 | 21,371 | 21,371 | 17,598 |
| MiBench | Guthaus et al. (2001) | MIT | 40 | 40 | 38 |
| OpenCV | OpenCV | BSD 3-Clause | 442 | 442 | 254 |
| POJ104 | Mou et al. (2015) | MIT | 49,816 | 49,815 | 49,804 |
| stencil_synth | Ben-Nun et al. (2018) | BSD 3-Clause | 12,800 | 12,800 | 12,721 |
| Tensorflow | Abadi et al. (2015) | Apache 2.0 | 1,985 | 1,985 | 683 |
| **Total** | | | **117,967** | **117,922** | **103,813** |

## A  VOCABULARY AND DATASET ANALYSIS

In order to create the vocabulary, we collected a large, heterogeneous collection of compilable C/C++ and LLVM-IR code sourced from a wide range of open-source projects and publicly available benchmarks. This dataset includes over 100 k code samples, covering libraries for scientific computation, biologically-oriented projects, and executable code from popular GitHub repositories. A summary of the sources for the code samples is provided in Table 3.

We converted the code samples to graphs by either compiling the source code from scratch using Clang (Clang), adapting the compilation procedure for each subset, or by downloading pre-compiled LLMV-IR code files from available collections, such as the CompilerGym project (Cummins et al., 2021a). The ProGraML Python library (Cummins et al., 2021b) was then used to generate graph representations. We discarded source code files that resulted in compilation errors and LLVM-IR files that took longer than five seconds to convert into graphs. Moreover, to ensure meaningful samples and stabilize the training process considering memory constraints, we excluded graphs with fewer than 5 or more than 3,000 nodes. These thresholds were selected on the basis of the distribution of unfiltered graph nodes, ensuring that not more than 10% of the graphs would be removed.

After generating all graphs, we found that 99.5% of the nodes in the dataset could be represented with a dictionary of only 341 tokens. We included the `[CLS]` and `[META]` tokens to represent the corresponding nodes in our graph representation, and an additional `[UNK]` token, to map all infrequent elements of the language, bringing the total size of the set to 344. This set is sufficiently general, covering a significant portion of the nodes from other datasets as well: DevMap has only 4.97% of nodes not covered by the vocabulary, while POJ-104 has 1.05% and ExeBench has 0.55%. Therefore, we employ this collection of tokens as our main vocabulary for all experiments.

