# OpenReview forum: "GEPCode: A Context-Aware 1M-Parameters Graph-Based Language Model for Source Code"
_ICLR.cc/2025/Conference — Submitted to ICLR 2025_

### Official Review · Reviewer_jpTT · 2024-10-18

**Soundness:** 2
**Presentation:** 3
**Contribution:** 2
**Rating:** 5
**Confidence:** 4

**Summary:**

The paper proposes GEPCode, a Graph Neural Network (GNN) to embed compiler intermediate representation of source code. GEPCode is pre-trained on 1.3M graphs constructed from source code for three objectives: (1) Attribute Masking, (2) Meta Prediction, and (3) Contrastive Learning. The resulting model is then fine-tuned for two downstream tasks respectively: (1) Device Mapping and (2) Algorithm Classification. Evaluation shows that GEPCode, which has only 1M parameters, achieves comparable performance with respect to much larger pre-trained transformer models, highlighting its efficiency.

**Strengths:**

* While previous works have approached the same problem with graph representations, GEPCode is the first to *pre-train* an GNN for intermediate representation, and proposed the three new objectives for pre-training.
* The proposed method is compared against a comprehensive list of baseline methods. All experiments are repeated 5 times for reliability.
* The paper is written clearly.

**Weaknesses:**

* The proposed method does not show a clear advantage over Perfograph which performs better on DevMap with an even smaller model size, and does not require pre-training.
* It is highlighted that GEPCode is more parameter-efficient than pre-trained transformers. I do not think this is a well-established advantage of the proposed method unless it can be demonstrated that GEPCode outperforms the SOTA pre-trained transformer of the same model size.
* Regarding efficiency, I'm not sure about the practical benefit of reducing the inference time from around 100ms (with CodeT5 or other transformers) to 23ms (with GEPCode), at the cost of accuracy degradation. Those transformers are already pretty small and fast, whose inference latency should be acceptable for DevMap and algorithm classification that are considered in evaluation.
* The baseline transformer models are mostly trained on multilingual programming language data, while GEPCode is trained on IR which is a single language. Is it possible that a monolingual transformer model trained only on the target language for the evaluation task would perform better, and thus the baseline numbers are underestimated?

**Questions:**

* A transformer model over flattened source code is naturally a GNN over fully connected graphs. It would be interesting to study how it performs to directly pre-train a transformer model over the same data with the same objectives.
* L292 is confusing to me. Is the "normalized property value" in META edges the same as $m$ in eq.6? If that's the case, isn't the training objective already known from the input?

---

> ### Author Response · Authors · 2024-12-03
> **Answers**
>
> Q1: A transformer model over flattened source code is naturally a GNN over fully connected graphs. It would be interesting to study how it performs to directly pre-train a transformer model over the same data with the same objectives.
>
> A1: Yes, transformer models identify relationships across the entire context, effectively constructing a graph of relationships from a complete graph. However, our approach aims to leverage domain-specific knowledge about the structure of code. This goes beyond merely considering the sequence of op-codes; we focus on capturing the complex dependencies in data and control flow. By doing so, we aim to build a more efficient and lightweight model. To improve the comparison in the future, we will train a transformer model under the same conditions as the GNN, as suggested by the reviewer.
>
> Q2: L292 is confusing to me. Is the "normalized property value" in META edges the same as m in eq.6? If that's the case, isn't the training objective already known from the input?
>
> A2: Yes, the classification targets (the META properties) are present in the input as nodes in the initial graph. However, the meta prediction is performed on the final representation of the CLS node, rather than directly on the representations of the META edges. Its purpose is simply to guide the CLS node to pay greater attention to the META nodes, as explained at the beginning of the paragraph: "To enhance the network’s ability to effectively capture meta-information into the global representation, ...". This allows to include meta nodes in a structural way for graph allowing generalization for downstream tasks.

---

### Official Review · Reviewer_V1oP · 2024-11-01

**Soundness:** 3
**Presentation:** 3
**Contribution:** 2
**Rating:** 3
**Confidence:** 4

**Summary:**

This paper introduces GEPCode, a novel approach using a Graph Neural Network to model compiler intermediate representations of code with the adaptability to accommodate new tasks. The model is first pre-trained as a general-purpose language model, then fine-tuned on downstream tasks, achieving results comparable to state-of-the-art models trained from scratch. Additionally, GEPCode performs similarly to leading pre-trained Transformer-based language models while requiring fewer parameters, offering a more flexible and efficient solution.

**Strengths:**

The problem statement addressed in this paper is worth solving as there is demand for smaller, efficient and flexible models in this domain.

**Weaknesses:**

(1) Typically, fine-tuning is performed for 1 to 10 epochs. Here, fine-tuning for both downstream tasks was run for 100 epochs, which could have led to overfitting.

(2) The accuracy of fine-tuned GEPCode on downstream tasks is compared against pre-trained transformer-based language models. A comparison with fine-tuned transformer-based language models for our downstream tasks would have been ideal.

**Questions:**

Please clarify if IRGen is a non-transformer-based pre-trained language model that performs better than GEPCode.

---

> ### Author Response · Authors · 2024-12-03
> **Answers**
>
> Q1: Please clarify if IRGen is a non-transformer-based pre-trained language model that performs better than GEPCode.
>
> A1: IRGen is not a competitor of our model, it is a framework based on genetic algorithms to identify sequences of optimization flags that can significantly improve embedding quality. It is thus an orthogonal research to the one presented in this paper that can be hybridized to further improve the embedding quality in future works.

---

### Official Review · Reviewer_NyAs · 2024-11-05

**Soundness:** 3
**Presentation:** 3
**Contribution:** 2
**Rating:** 5
**Confidence:** 3

**Summary:**

The paper introduces GEPCode, a graph-based, efficient, pre-trained, context-aware LM of graph representations of source code. The proposed approach expands upon ProGraML and Perfograph. GEPCode expresses LLMV-IR code samples as graphs, where nodes are token within a vocabulary of LLVM-IR elements, and edges are directed and represent dependencies between the elements of code. GEPCode is pretrained using synth-compilable subset of the Exebench dataset, using three tasks - attribute masking, meta prediction, and contrastive learning. For downstrem tasks, GEPCode is evaluated on device mapping and algorithm classification. Experimental results demonstrate that GEPCode is able to bridge the gap between the efficiency of task specific architectures and the generality of larger LMs, while using a limited number of parameters.

**Strengths:**

- Overall the paper is written well. The motivation behind the model choice is sound.
- The pretraining objectives that include three tasks are effective.
- Experiment results show the value of the proposed method.

**Weaknesses:**

- The biggest weakness of the work is the novelty. In a nutshell, it is a paper that is using GNN for some specific code tasks. GNN was previously explored for coding tasks, however, with the emergence of LLMs, the focus has shifted. In this paper, authors emphasized on compact LMs, but why GNN is the solution, it is not clear.
- The paper could use more software engineering tasks for evaluation. In the analysis part of the work, there is not much critical thinking paid by the authors. Straight-forward main results and a piece of ablation study - that's it. Moreover, I didn't understand the baselines used in comparison.

**Questions:**

- Why GNN is used as a solution to build compact LMs in this work? What is the motivation.
- The paper uses LLMV-IR representation of code; in that case, is the proposed method scalable?

---

> ### Author Response · Authors · 2024-12-03
> **Answers**
>
> Q1: Why GNN is used as a solution to build compact LMs in this work? What is the motivation.
>
> A1: Graph Neural Networks are used as a solution to build compact language models because of their ability to effectively capture and process structured relational data. Source Code inherently has a graph-like structure and GNNs can represent and process such structures more naturally and efficiently encoding domain-specific relationships enhancing the model’s understanding without significantly increasing complexity. Moreover GNNs allow us to include meta node population in a structural way allowing generalization in downstream tasks.
>
> Q2: The paper uses LLMV-IR representation of code; in that case, is the proposed method scalable?
>
> A2: LLVM-IR captures fine-grained operations, its verbosity could lead to scalability challenges addressable simplifying the LLVM-IR by removing irrelevant nodes or edges or splitting large graphs into smaller ones. At the same time, however, the use of LLVM enables several advantages including the application of the methodology to different high-level languages in an agnostic manner and the extension of the methodology to different domain-specific dialects.

---

### Meta-Review · Area_Chair_2Qt3 · 2024-12-23

**Metareview:**

This paper proposes GEPCode, a GNN-based pre-trained model over LLVM-IR. It uses three pre-training objectives—attribute masking, meta prediction, and contrastive learning—and is tested on device mapping and algorithm classification. The authors claim GEPCode outperforms larger transformer-based models in efficiency while maintaining comparable accuracy.

**Strengths:**

* The paper highlights “demand for smaller, efficient and flexible models in this domain” (V1oP), which is timely given the growing focus on efficiency in modern ML research.

* Multiple reviewers (e.g., NyAs, jpTT, V1oP) note that the paper is “written well” (NyAs) and “written clearly” (jpTT).

* The inclusion of three tasks—attribute masking, meta prediction, and contrastive learning—finding that these “are effective” (NyAs)

* Comprehensive set of experiments. (jpTT)

**Weaknesses:**

* Unclear motivation of using GNN: “why GNN is the solution, it is not clear” (NyAs).

* It is also unclear how the baselines were chosen (NyAs), and a comparison with fine-tuned transformer-based language models would be beneficial (V1oP)

* Weak comparison with prior methods. Perfograph performs better on DevMap with fewer parameters and no pre-training (jpTT).

Though the paper is well-written and tackles a relevant problem, reviewers remain unconvinced about GEPCode’s novelty and its empirical advantages. The authors’ rebuttal did not fully address these concerns.

**Additional Comments On Reviewer Discussion:**

Please refer to the meta-review for details.

---

### Decision · Program_Chairs · 2025-01-22

Reject